# New Insights on the Role of Connexins and Gap Junctions Channels in Adipose Tissue and Obesity

**DOI:** 10.3390/ijms222212145

**Published:** 2021-11-10

**Authors:** Jorge Enrique González-Casanova, Samuel Durán-Agüero, Nelson Javier Caro-Fuentes, Maria Elena Gamboa-Arancibia, Tamara Bruna, Valmore Bermúdez, Diana Marcela Rojas-Gómez

**Affiliations:** 1Facultad de Ciencias de la Salud, Instituto de Ciencias Biomédicas, Universidad Autónoma de Chile, Santiago 8910060, Chile; jorge.gonzalez@uautonoma.cl (J.E.G.-C.); Nelson.caro@uautonoma.cl (N.J.C.-F.); 2Facultad de Ciencias Para el Cuidado de la Salud, Universidad San Sebastián, Sede Los Leones, Lota 2465, Providencia, Santiago 7500000, Chile; samuel.duran@uss.cl; 3Facultad de Química y Biología, Universidad de Santiago de Chile, Av. Libertador Bernardo O’higgins 3363, Estación Central, Santiago 9170022, Chile; maria.gamboa.a@usach.cl; 4Centro de Investigación Austral Biotech, Facultad de Ciencias, Universidad Santo Tomás, Avenida Ejercito 146, Santiago 8320000, Chile; tbruna@australbiotech.cl; 5Facultad de Ciencias de la Salud, Universidad Simón Bolívar, Barranquilla 080002, Colombia; v.bermudez@unisimonbolivar.edu.co; 6Escuela de Nutrición y Dietética, Facultad de Medicina, Universidad Andres Bello, Santiago 8370321, Chile

**Keywords:** connexins, gap junctions channels, adipose tissue, obesity, cardiovascular diseases

## Abstract

Due to the inability to curb the excessive increase in the prevalence of obesity and overweight, it is necessary to comprehend in more detail the factors involved in the pathophysiology and to appreciate more clearly the biochemical and molecular mechanisms of obesity. Thus, understanding the biological regulation of adipose tissue is of fundamental relevance. Connexin, a protein that forms intercellular membrane channels of gap junctions and unopposed hemichannels, plays a key role in adipogenesis and in the maintenance of adipose tissue homeostasis. The expression and function of Connexin 43 (Cx43) during the different stages of the adipogenesis are differentially regulated. Moreover, it has been shown that cell–cell communication decreases dramatically upon differentiation into adipocytes. Furthermore, inhibition of Cx43 degradation or constitutive overexpression of Cx43 blocks adipocyte differentiation. In the first events of adipogenesis, the connexin is highly phosphorylated, which is likely associated with enhanced Gap Junction (GJ) communication. In an intermediate state of adipocyte differentiation, Cx43 phosphorylation decreases, as it is displaced from the membrane and degraded through the proteasome; thus, Cx43 total protein is reduced. Cx is involved in cardiac disease as well as in obesity-related cardiovascular diseases. Different studies suggest that obesity together with a high-fat diet are related to the production of remodeling factors associated with expression and distribution of Cx43 in the atrium.

## 1. Introduction

Obesity and overweight are defined as an excessive or abnormal accumulation of adipose tissue with health-risk consequences. Obesity prevalence has approximately tripled in the past three decades with a mortality rate of 2.8 million per year associated with this disease [1,2,3]. The excessive increase in obesity prevalence has been impossible to halt; therefore, it is necessary to discern in more detail the factors involved in its pathophysiology to more clearly understand the biochemical and molecular mechanisms comprised in adipocyte formation and development, as well as adipose tissue behavior.

Adipose tissue is composed of different cell lineages, consisting mainly of adipocytes, which are specialized cells that store energy reserves in the form of triglycerides. However, other cell types make part of this tissue, including fibroblasts, adipocyte precursors, endothelial cells, and immune cells such as macrophages, natural killer cells, T helper lymphocytes, regulatory T lymphocytes, and B lymphocytes [4,5].

Adipose tissue is a metabolically and physiologically complex organ with functions that are not restricted merely to energy storage. It is an active tissue with hormonal, immunological, and energy homeostasis functions, where angiogenic processes also take place [4]. In the individual, adipose tissue has a high capacity of adaptation to different energy conditions, allowing this tissue to undergo continuous remodeling processes, including new fat-cell generation, known as adipogenesis.

In the process of adipogenic differentiation, after receiving a specific extracellular signal a mesenchymal cell of mesodermic origin initiates the process of differentiation into a pre-adipocyte. These signals allow a series of subsequent events to take place that ultimately result in the formation of a mature adipocyte. Therefore, differentiation includes morphological changes, cell growth arrest, and expression of specific lipogenesis-related proteins [6,7,8].

Adipocyte differentiation occurs throughout the life cycle of an individual, as a response to store additional energy in the form of fat or to restore cells lost by physiological aging processes. As in any other cell differentiation process, the molecular and biochemical mechanisms that regulate adipogenesis are highly orchestrated, where several transcription factors are involved in adipogenic regulation [9].

Adipogenesis comprises consecutive changes involving gene expression to achieve a mature adipocyte, including growth arrest, mitotic clonal expansion, and early and terminal differentiation [10,11].

In an initial stage of differentiation, cellular processes activate proteins belonging to the Activator protein 1 (AP1) family of transcription factors, which leads to the expression of the receptor activated by Peroxisome proliferator-activated receptor gamma (PPARγ). This transcription factor is considered the master regulator of adipogenesis, since it regulates the expression of a large number of genes related to cell differentiation and the accumulation of lipids in the cell [12,13,14,15,16].

Other additional factors contributing to mature adipocyte formation are the signal transducers and activators of transcription (STATs), and member proteins of the CCAAT/enhancer-binding proteins (C/EBP), which fulfill essential functions during adipogenesis [12,17]. In addition to positive adipogenesis regulators, important and potent negative regulators have been described, which include proteins from the GATA and WNT family [18,19,20,21,22,23,24].

## 2. Connexin Forms Gap Junction Channels and Hemichannels

The ability of animals to adapt to variable environmental conditions, adopting and transitioning between different phenotypes, is mediated by efficient communication and a synchronized response. In vertebrates, cell communication can be indirect or direct. In the former, cellular communication comprises neuronal and hormonal signaling mechanisms between distant cells, as well as local signaling between adjacent cells. Auto- and paracrine mechanisms allow a coordinated function of tissues and organs. On the other hand, in direct communication, interaction between cells is mainly carried out by Gap Junctions (GJ). This type of communication corresponds to a specialized type of connection or channel between neighboring cells with the ability to open or close, thus providing a direct and selective conduit between their cytoplasms. The concept of GJ was established in 1967 by Jean-Paul Revel and Morris Karnovsky [25], who were the first to describe the presence of these intercellular junctions.

Gap junctions are formed by protein subunits called Connexins (Cxs), where six Cxs form a hemichannel (HC) (or connexon). Gap Junction Channel (GJC) structure comprises the serial coupling of two hemichannels, one provided by each cell. Connexins have cytosolic amino-terminal and carboxy-terminal ends, one cytoplasmic loop (CL), four transmembrane domains called M1, M2, M3 and M4, and two extracellular loop domains (EL1 and EL2) [26,27,28,29]. The Cx has an average size of 380 amino acids, and the most commonly used connexin classification system is based on their molecular weight (Figure 1) [30,31].

Twenty-one different types of Cxs have been described in the human and mouse genome, in addition to an increasing number of orthologs in other vertebrates [32]. Hence, in the literature, they are identified by Cx, the abbreviation of Connexin, followed by a number corresponding to the estimated molecular weight, for example, Cx43 is a protein of 43 kDa. Of the different identified connexins, Cx43 is essentially expressed in all cell types. Consequently, the wide variety of processes and functions in which it is involved has been subject of research.

It has been described that GJs are present in almost all tissues, including nerve and muscle tissue, in the liver, and in the retina, where they participate in a wide variety of processes, such as embryonic development [33], wound healing and cell differentiation [34]. Additionally, GJs are the basis of electrical coupling in excitable cells such as neurons and heart cells, mediating action potential propagation [35]. Alterations in the regulation of their expression and their cellular distribution are related to different diseases [36], for instance tumors [37], epilepsy [38], atherosclerosis [39], and heart diseases [40]. Gap junctions allow the coordinated transport of small molecules, such as ions, amino acids, nucleotides, second messengers (Ca^2+^, cAMP, cGMP, IP3), and various metabolites such as ADP, glucose, lactate, and glutamate [41].

Post-translational phosphorylation of Cxs plays a key role in HC and GJC function, altering hydrophobicity and even the structure of the protein that forms the channel. These modifications can influence channel activity or may change Cx interaction with other proteins [42,43,44,45]. Connexin phosphorylation occurs through serine/threonine kinases or tyrosine kinases, such as PKC, MAPK, CaMKII, casein kinase or PKA, among others. Phosphorylation performed by PKC is related to a decrease in GJC activity [46,47,48]. In contrast, in Cx43 AKT-dependent phosphorylation increases the GJ’s size and its activity [49,50]. Consequently, this phosphorylation of Cx43 by AKT promotes transition of Cx43 HC at the periphery of the GJ plaque in to the plaque by releasing the interaction with ZO-1 [50,51]. Calmodulin-dependent phosphorylation results in a decreased Gap junction activity [52,53,54]. MAPK phosphorylation is related to a rapid internalization and inhibition of the connexin channel activity [47,55]. Collectively, Cx phosphorylation is associated with changes in activity, assembly, and stability.

GJCs allow the exchange of cytoplasmic molecules with size up to ~1 kDa between coupled cells, while HCs constitute a means of paracrine communication and provide molecule exchange between the extracellular milieu and the cytoplasm [56,57,58,59]. Under physiological conditions, HCs are preferably closed to preserve ion homeostasis [60]. Since Cx HCs form poorly selective ion channels of high conductance, their opening leads to an influx of Ca^2+^, breakdown of the electrochemical gradient across the plasma membrane, and loss of essential metabolites. HCs are activated under pathological conditions, including oxidative stress, mechanical stretch, inflammatory processes, and low pH [61,62,63,64,65,66,67]. An exacerbated HC activity during pathological states can increase cell damage. Excessive release of ATP or glutamate is an indicator of toxicity to neighboring cells and can spread the damage to distant cells [68,69,70].

## 3. Connexins, Gap Junctional Communication, Hemi-Channels

### Association with Adipose Tissue

Adipose tissue is of mesodermal mesenchymal origin, consisting mainly of adipocytes [6]. There are two main classes of adipose tissue: white adipose tissue (WAT) and brown adipose tissue (BAT) [71,72]. Both tissues differ in morphology, function, and are molecularly distinct among others [73]. White adipocytes specialize in storing energy, with a morphological feature of a peripheral nucleus and a single lipid droplet. Contrarily, brown adipocytes, whose function is to dissipate energy and regulate heat production, have a central nucleus, and in its cytoplasm small drops of lipids are contained.

During adipogenesis, it is necessary for *Cx43* expression and activity to starkly decrease. Preliminary studies by the Azarnia group [74] described a progressive loss of GJC activity during adipogenesis in mouse 3T3-L1 cell line fibroblast cultures.

Umezawa et al. 1992 [75] showed that expression of Cx43 is downregulated at the transcriptional level during adipocyte differentiation of H-1/A marrow stromal cells. The role of Cx in the early stages of adipogenesis was analyzed by the Yanagiya group [76]. Their data showed that the characteristic increase in DNA synthesis and the number of cells, attributed to the initial stage of differentiation, were inhibited by the presence of the GJ blocker 18-α-glycyrrhetinic acid (AGRA), thus indirectly demonstrating that GJCs are essential for mitotic clonal expansion during adipogenesis.

In a more detailed study of Cx43 during the different stages of adipogenesis conducted by Yeganeh et al. [77], it was demonstrated that during the early stages of differentiation, Cx43 was strongly phosphorylated; additionally, it was translocated from the endoplasmic reticulum to the plasma membrane. In the intermediate and late stages of differentiation, Cx43 phosphorylation levels decreased and Cx43 was removed from the cell membrane to be degraded in the proteasome. Additional experiments also established that inhibition of Cx degradation by the proteasome resulted in the arrest of adipogenic differentiation (Figure 2).

On the other hand, Cx43 plays a role in mesenchymal cell fate. Yamanouchi et al. [78] demonstrated that Cx43 inhibition in skeletal muscle cell culture favored changes in phenotype, promoting triglyceride accumulation and C/EBPα expression. In their study, in muscle cells exposed to differentiation medium AGRA treatment did not have an effect on adipogenesis, giving rise to mature adipocytes.

Additionally, Schiller and collaborators demonstrated that, in cultures of murine osteogenic cell line MC3T3-E1, inhibition of Cx43 by AGRA or oleamide halted the maturation of pre-osteoblastic cells and favored trans-differentiation of osteoblasts into adipocytes [79]. These events were concomitant with an increase in lipoprotein lipase and PPARγ expression [79].

More recently Chen et al. [80] related Cx43 activity and capsaicin-mediated lipolysis. This group demonstrated that activation of the transient receptor potential V1 (TRPV1) by capsaicin increased the influence of calcium in 3T3-L1, augmenting lipolysis. This effect was counteracted by the use of AGRA, concluding that Cx43 activity plays an intermediary role in lipolysis. Experiments in mice fed high-fat diets and exposed to dietary capsaicin had less accumulation of perirenal, mesenteric, and testicular adipose tissue and an increase in *Cx43* expression in this tissue [80]. A recent study by Turovsky et al. [81] demonstrated a role of Cx43 HC in the stimulation of lipolysis in white adipocytes.

Most studies exploring the role of Cx43 in adipogenesis have been carried out on mesenchymal cells with differentiation capacity towards osteogenic, chondrogenic, and adipogenic lineage. However, the group of Shao et al. studied Cx43′s role in adipogenesis differentiation in human-induced pluripotent stem cells (iPSCs). They employed wild-type iPSCs cell culture and iPSCs carrying a *GJA1* (Cx43) gene mutation from patients suffering from oculodentodigital dysplasia, which were able to differentiate into adipocytes. Interestingly, in their study, *Cx43* expression was increased during differentiation. To clarify the role of Cx43 during differentiation, the researchers used iPSC cell cultures carrying complete *GJA1* (Cx43) genetic ablation and observed that this cell type maintained the ability to differentiate towards the adipocyte phenotype. However, Cx43-ablated cells showed a higher propensity to suffer premature senescence when compared to control cells. Their data demonstrated that GJ proteins can play a role in multipotent stem cells’ self-renewal [82].

In addition, Mannino et al. [83] reported a decrease of *Cx* expression (Cx43, Cx32 and Cx31.9) during adipogenic differentiation of adipose-derived stem cells. These authors suggest that the presence of multiple Cx isoforms is in line with a high initial differentiation ability of adipose-derived stem cells that is markedly restricted during adipogenic differentiation.

The role of Cx43 is currently under study in pathologies related to adipogenesis. It has been reported that transgenic mice carrying a Cx43 mutation -G60S- presented an early osteopenia phenotype with a significant increase in bone marrow adiposity, which was in parallel accompanied by a reduction in GJC formation and function. This increase in adipogenic differentiation was accompanied by PPARγ activation [84].

Brown adipose tissue (BAT) is an organ specialized in regulating body temperature, particularly in humans in the neonatal period. Only mammals possess this tissue, and it is responsible for producing heat when the body temperature is below normal physiological levels. Brown adipose tissue thermogenic activity is directed by the sympathetic nervous system and by the mitochondrial uncoupling protein 1 (UCP1), which decouples the ATP production to produce heat [6,71].

Brown and white adipose tissue have a common stem-cell mesodermal precursor. However, during embryonic development, instructive signals establish their different phenotypes [6]. Such is the case for Cx43, which is higher in BAT than in WAT. In an effort to understand the role of Cx in BAT, studies by Zhu et al. demonstrated that Cx43 plays a role during the “beiging” process of white fat in adipose tissue. “Beige” adipocytes are present in white adipose tissue and dissipate energy as heat as described in Zhu et al. [85]. Beiging of white adipose tissue was suggested as a therapeutic strategy for weight loss in humans [86]. Beige adipocytes residing in mice WAT have greater cell–cell communication via GJC when compared with the intercellular communication established by GJC in “white” adipocytes [85].

Under cold induction when a white adipocyte has the potential to become beige, it requires *Cx43* expression and activity in a cAMP coupling dependent manner, via activation of β3-adrenoceptor [85]. This was verified by experiments involving *Cx43* gene downregulation or by AGRA pharmacological inhibition of this protein, which ultimately resulted in reduction of neuronal activation-induced beiging. Moreover, overexpression of Cx43 in mice exposed to mild cold stress was sufficient to induce WAT beiging [85].

On the other hand, in Chagas disease, adipose tissue is a relevant cell target for *Trypanosoma cruzi* parasite infection. A characteristic of this disease is GJC expression and activity alteration. Results of experiments with mice infected with *T. cruzi* demonstrated an alteration in the expression and protein activity of GJC in BAT adipocytes, establishing a decrease in BAT *Cx43* expression and protein activity. However, in WAT an opposite effect was observed, increasing the levels of Cx43 and GJC activity [87].

The brown color of BAT tissue is due to the high content of highly branched vascularization and numerous mitochondria [88]. The presence of Cx in the mitochondrial membrane (mtCx) has been described in diverse cell types, including BAT [89,90].

Kim et at. [89] studied the role of Cx43 in the functioning of mitochondria in BAT fat cells, beyond the classical role of Cx43 as GJC. They used an animal model of adipocyte-specific Cx43 knockout (*Gja1 ^adipoq^KO*) subjected to cold stress and a high-fat diet. The BAT of these mice showed a lower presence of mitochondria, along with increased mitochondrial damage. The *Gja1 ^adipoq^KO* mice exposed to β3-adrenoceptor activation by the CL316,243 agonist exhibited reduced mitochondrial integrity [89] (Figure 3).

## 4. Obesity, Atrial Fibrillation and Cx

Different studies have shown the intimate relationship between obesity and atrial fibrillation (AF) [91,92]. Atrial fibrillation is the most common type of heart arrhythmia, where the central feature is very rapid and uncoordinated atrial activity [39]. A key feature of AF is atrial enlargement and remodeling in the expression, function, and location of cardiac Cxs [93,94,95]. Under normal conditions, GJCs are located at the intercalated disc of the cardiomyocyte and play a role in action potential propagation [96]. Hence, any alteration in the expression, function or location of GJC contributes to the formation of an arrhythmogenic substrate [97,98,99]. Collectively, the arrhythmia caused by Cx remodeling is considered the fundamental cause of AF.

In the heart, adipose tissue located in the epicardial and pericardial areas makes part of its anatomy [100,101,102]. In individuals affected by obesity, fatty tissue can represent up to 20% of the cardiac mass, where cardiac steatosis may develop. It has been observed that adipose tissue participates in the pathophysiology of AF, as it can contribute to tissue remodeling, including increases in fibrosis and fatty infiltration [103,104,105]. Furthermore, pericardial and epicardial fat deposition results in altered conduction, where atria are structurally and electrically remodeled, inducing AF [106,107,108].

Cardiometabolic changes produced by fatty tissue reduce metabolic flexibility, resulting in oxidative stress, inflammation, and fibrosis [108,109,110,111]. In patients with AF, high levels of pro-inflammatory molecules in adipose tissue of the pericardium have been observed [112,113]. Cytokines produced in adipose tissue or paracrine factors secreted during the inflammation process might reach the atrium, and prompt further structural and electrical remodeling. Various cytokines have been linked to arrhythmia, including TNF-alpha, Interleukin-6, Interleukin-8 and Interleukin-10 (IL10). Kondo et al. [114] demonstrated that IL10 treatment ameliorates high-fat diet-induced inflammatory atrial remodeling and fibrillation [114]. Systemic inflammation has been hypothesized to promote atrial electric remodeling as a result of cytokine-mediated changes in connexin expression [115].

On the other hand, in addition to cardiac fat, visceral fat depot is a source of free fatty acids and bioactive molecules such as adiponectin, resistin, and inflammatory cytokines, factors that may participate in AF pathogenesis [116,117]. Free fatty acid overload in patients with obesity induces lipid accumulation within cardiomyocytes and apoptosis, which might also trigger inflammation [118].

Based on the relationship between adipose tissue components and the presence of AF, various groups have investigated specific factors that regulate atrial electrical remodeling, and the associations between adipose tissue and regulation of *Cx43* expression and/or activity during AF [119,120].

The state of obesity can affect cardiac *Cxs* expression and protein phosphorylation, and in addition, alter Cx degradation, location, and activity [105,117]. Moreover, in the physiological environment surrounding an organism with obesity, the oxidative and pro-inflammatory stress processes characteristic of adipose tissue can influence cardiac GJ protein abnormalities [117].

More recently, Sato et al. [120] studied the relationship between cardiac steatosis and AF in mice overexpressing *Perilipin 2* (*PLIN2*). In their study, transgenic mice showed increased atrial fat accumulation and electrocardiographic abnormalities, with a higher prevalence of persistent AF. In these mice, cellular Cx43 distribution was altered. Contrary to the location at the intercalated disc in cardiomyocyte of wild-type mice, Cx43 was heterogeneously and laterally distributed in atrial cardiomyocytes of mice overexpressing *PLIN2*.

Concerning human studies, immunohistochemical analysis of the right atrial tissue from patients with sinus rhythm or AF showed a correlation between BMI and Cx43 remodeling in atrial tissues. Patients affected by obesity had an increased presence of Cx43 in the lateral position of the atrial cell compared to lean patients [121].

## 5. Atrial Fibrillation, Cx and High-Fat Diets

Different studies suggest that obesity and high-fat diets (HFD) are related to the production of remodeling factors associated with expression and distribution of *Cx43* in the atrium [119,122,123,124].

Experiments by Meng et al. [124] with HFD-fed rats observed obesity in one-third of the animals along with an alteration in lipid homeostasis. The weight of the atrium was higher in HFD and non-obese HFD rats in comparison with rats fed with a normal diet. Additionally, the remodeling of the GJC was evidenced by changes in the expression as in the cellular location of Cx43 and Cx40. Moreover, Takahashi et al. demonstrated that HFD mice have reduced expression of Cx40 (mRNA and protein) and lateralization of Cx40 in the atria [125].

Zhong et al. [126] conducted experiments with APOE^−/−^ mice fed with HFD and demonstrated a greater susceptibility to cardiac arrhythmias and electrical remodeling, with an increase in cardiac expression of *Cx43*. The presence of obesity/hyperlipidemia was associated with an increase in *CaMKII* expression. Treatment with CaMKII inhibitor, KN93, reduced the slow cardiac conduction, and *Cx43* expression levels were nominalized.

The influence of high-fat consumption independent of body weight on cardiac *Cx* expression and activity was evidenced in experiments using a murine model with HFD that did not develop obesity, hyperlipidemia or hyperglycemia [123]. The results of this work demonstrated the presence of an arrhythmic phenotype, with a greater susceptibility to ventricular tachyarrhythmias, which was accompanied by a decrease in Cx43 phosphorylation and an increase in Cx43 lateralization in the cardiomyocyte [123].

Similar studies by Jin et al. [127] in mice fed with HFD showed increased blood glucose, body weight, total cholesterol, triglycerides, hemoglobin A1c, insulin, and brain natriuretic peptide. In addition, HFD significantly down-regulated *Cx43* and upregulated *β-catenin*, *N-cadherin*, and *plakoglobin* in the hearts of HFD-fed mice compared with mice fed with a normal diet. Long-term HFD in mice resulted in left ventricular hypertrophy, interstitial fibrosis, and dysregulation of renin-angiotensin-system (RAS). It was shown that HFD produced cardiac remodeling and change in interstitial collagen expression through RAS activation.

In another study, obesity and metabolic syndrome were induced in rats with high sucrose diets (HSD). In these experiments, the animals showed an increase in serum cholesterol and TG levels, in body weight, heart weight and in amount of retroperitoneal and epicardial fat. In addition, Cx43 and PKC signaling cascades were altered in the myocardium. Downregulation of cardiac *Cx43* and a decrease in phosphorylated Cx43 were related to increased predisposition to ventricular arrhythmias [48].

Concerning therapies that seek to mitigate the deleterious effects of HFD or High Fat Fructose (HFF) deleterious effects on action potential propagation in the heart, it has been shown that different supplements or compounds can counteract arrhythmias. Perdicaro et al. [128] demonstrated in Wistar rats fed with HFF that grape pomace extract prevented diet-induced heart alterations. Moreover, this effect was accompanied by a decrease in the non-phosphorylated form of Cx43.

On the other hand, the deleterious effects of HFD, such as decreased *Cx43* expression and lateralization, were partially reversed using melatonin and omega-3 polyunsaturated fatty acids (PUFA). Intake of these compounds increased *Cx43* expression and its protein phosphorylation in atrial tissue [48]. Baum et al. showed omega-3 fatty acid inhibition of inflammatory cytokine-mediated Cx43 regulation in the heart [129].

In another study, liraglutide effects, an analog of glucagon-like peptide-1, on heart pathophysiology were evaluated in HFD-fed mice. Mice developed obesity, dyslipidemia and insulin resistance. Liraglutide treatment significantly decreased HFD-induced alterations related to glucose metabolism. The results also revealed that liraglutide restored Cx43 obesity-associated altered levels, without significant changes in animal body weight [130].

Another interesting aspect that has been recently studied is the role of Cx43 in the development of endoplasmic reticulum (ER) stress [131]. The ER is involved in protein synthesis and intracellular calcium storage. However, under excess body weight conditions and in comorbidities associated with this pathology, such as heart disease, hepatosteatosis, or obesity-associated insulin resistance, an increase in misfolded proteins or Ca^2+^ dysregulation results in an increase in ER stress in the different tissues [132]. In this context, recent works by Tirosh et al. [133] established that Cx43 intervenes in the propagation of the signals produced during ER stress in a mouse model exposed to HFD. The results of these experiments demonstrated an increase in the expression of Cx43 and GJ activity in the liver of obese animals, which was related to the transmission of ER stress. Likewise, the deletion of Cx43 in liver tissue decreased the ER stress caused by HFD.

Lastly, drug development targeting GJC is another aspect that has been studied in regard to Cx and obesity. Sasaki et al. [134] analyzed the behavior towards food intake in mice fed with HFD with and without high content of saturated fatty acids (SFA) and very high content of SFAs. The authors demonstrated that the application of INI-0602, an HC inhibitor, before HFD with very high SFA, prevented the intrinsic feeding rhythm alteration caused by high-fat diets. These findings indicate that Cx HCs might be involved in the molecular mechanisms underlying feeding pattern disturbance with HFD.

## 6. Conclusions

While the classic role of Cxs has been electrical coordination between excitable cells of the nervous or cardiovascular system, the function in other tissues remains to be elucidated in more detail.

In adipose tissue, Cx43 is involved in adipogenesis regulation and homeostasis maintenance. Increased *Cx43* expression at early stages and its subsequent degradation during intermediate and late stages are essential to ensure pre-adipocyte differentiation into a mature adipocyte phenotype. In brown adipose tissue development and regulation, Cx43 is essential in neuronal signaling for mitochondrial integrity preservation in BAT generation.

Due to the wide Cx distribution in different tissues, further studies are required to elucidate the mechanisms governing pathophysiology of obesity and the role of Cx proteins therein. For example, the understanding of Cx-mediated transmission of endoplasmic reticulum stress and its relationship with obesity, or the interplay between insulin signaling, insulin resistance, and gap junction function, are topics of scientific interest that need to be investigated in greater depth. Hence, drugs targeting Cx can be developed aiming at fighting obesity and related pathologies, thereby allowing reduction of the growing burden on health systems worldwide.

## Figures and Tables

**Figure 1 ijms-22-12145-f001:**
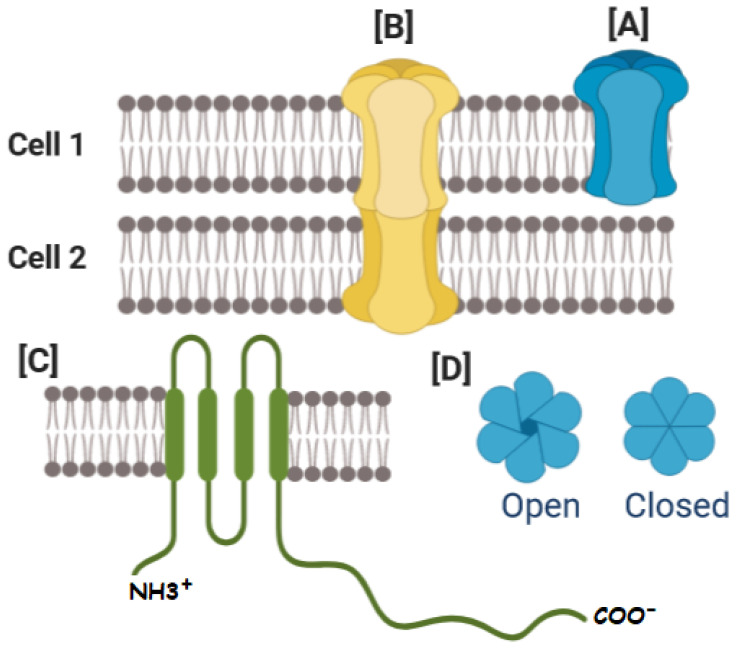
Connexins (Cxs), and connexin-based channels. (**A**) The oligomerization of six Cxs creates a hexamer named connexon or HC. (**B**) HCs interacting with opposing HCs from neighboring cells can dock to form different types of GJCs. (**C**) Cxs have cytosolic amino-terminal and carboxy-terminal ends, one cytoplasmic loop (CL), four transmembrane domains called M1, M2, M3 and M4, and two extracellular loop domains. (**D**) The size and charge of the aqueous pore formed by Cxs determine channel conductance and selectivity, allowing the cell to regulate various cellular processes, including the transport of molecules between connected cells. The gating of GJC indicated by the open-closed transition is modulated by a variety of factors, including voltage, H(+), Ca(2+), and phosphorylation.

**Figure 2 ijms-22-12145-f002:**
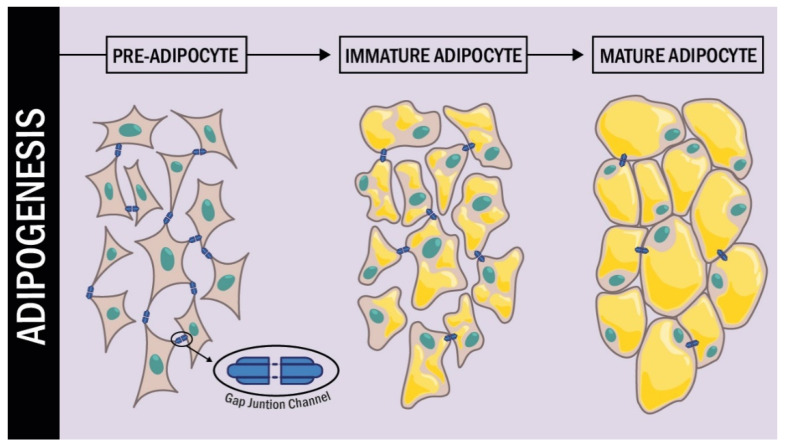
Cx43 expression during adipogenesis. During the process of differentiation from pre-adipocytes to mature adipocytes, the connexins undergo changes in their expression. Initially in a mesenchymal cell, the expression of Cx43 is high, but as differentiation occurs the expression levels of Cx43 decrease dramatically.

**Figure 3 ijms-22-12145-f003:**
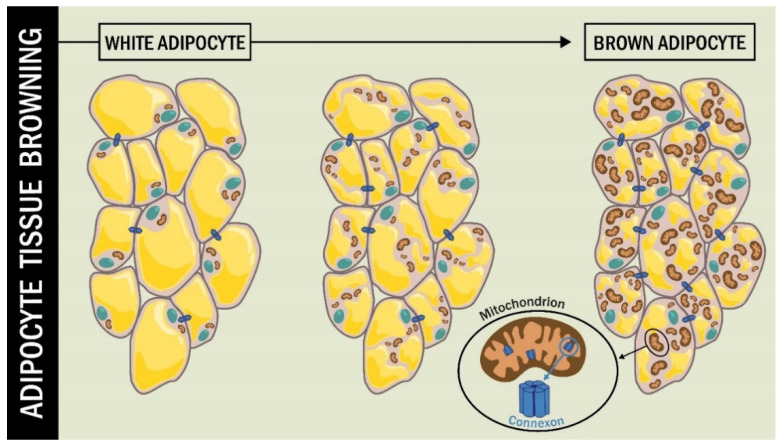
Cx43 during adipocyte tissue browning. During the browning process of WAT, the cell undergoes changes, including an increase in the number and size of mitochondria. In this process, the function of mtCx43 implies a protective role for mitochondrial integrity.

## Data Availability

Not applicable.

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
