# Peer review of "New Insights on the Role of Connexins and Gap Junctions Channels in Adipose Tissue and Obesity"

_ijms, 2021, doi:10.3390/ijms222212145_

Round 1

Reviewer 1 Report

Please see attached file for comments and suggestions

Reviewer 2 Report

Overall a nice review that places into the limelight the importance of gap junction function in the development and progression of obesity and its relation to cardiac disease. 

There are some issues that should be addressed:

  1. “Connexins, a family of related proteins, that form” would read better (line 24 page 1)
  2. Should not use un-introduced abbreviations in abstract (E.g. Cx43)
  3. "Moreover, it has been shown…." (Needs more detail) (lines 28-9)
  4. Abstract in general is poorly written and lacks fluidity and critical details, full of incomplete thoughts. Should be rewritten with a focus on clarity and completeness.
  5. “Cardiac disease obesity-related cardiovascular diseases” is redundant (line 34 page 1)
  6. "Angiogenic processes" or "angiogenesis" rather than "angiogenesis processes" (line 56-7 page 2)
  7. Inconsistent use of “indirect” and “direct” (lines 89-94 page 2)
  8. Cx43 should be expanded on as “for example Cx43 is a connexion of 43 kDa” (line 120-121 page 3)
  9. “Cx43, also known as GJalpha1,” has no context it should be introduced that there are two nomenclature systems or deleted. If two nomenclature systems are introduced it should be expected to explain that Cxs in different groups do not form heteromeric or heteortypic channels between groups (line 121 page 3)
  10. Tyrosine kinases (line 138 page 4)
  11. “In contrast in Cx43 AKT-dependent phosphorylation in- 140 creases the GJ’s size and its activity [49,50]. Consequently, this phosphorylation pre- 141 vents the interaction between Cx43 and the structural protein ZO-1 [50,51]” This is a little misleading, in reality phosphorylation of Cx43 by AKT promotes transition of cx43 HC at the periphery of the GJ plaque in to the plaque by releasing the interaction with ZO-1 rather than preventing it. (Line 140-142 page 4) See Sorgen et al 2018 (Int. J. Mol. Sci. 2018, 19(5), 1428; https://doi.org/10.3390/ijms19051428) for a comprehensive review of direct protein partner interactions of Cx43 and their regulation by phosphorylation.
  12. “Under pathological HCs are activated under conditions including oxidative stress, mechanical stretch, inflammatory processes, and low pH [61–67]. “ Poorly worded, perhaps should be "Under pathological conditions such as oxidative stress, mechanical stretch, inflammatory processes, and 153 low pH HCs are activated [61–67]."  (Line 152-3 page 4)
  13. “Interestingly, in their study, Cx43 expression was increased study during differentiation.” Poor English and confusing, I assume the word “study” in “study during” is extraneous (line 209 page 5)
  14. “Cx43 mutation -G60S mutation-“  the second “mutation is unnecessary (line 216 page 5)
  15. “The weight of the 322 atrium was higher in HDF and non-obese HDF mice in comparison with rats fed with 323 a normal diet.” Is this meant to be mice (line 322-3 page 8) If so should specify which study the rat study is being compared to 
  16. Since the primary focus of the review is Cxs in obesity and obesity related pathology, it would be of interest to include a section reviewing the interplay between insulin signaling, insulin resistance, diabetes, and gap junction functions since these are been studied rigorously in recent years. 

Round 2

Author Response

Thank you for this helpful review. Your corrections and suggestions were very valuable to obtain a better quality manuscript.

Reviewer 2 Report

The updated manuscript addresses all but one of this reviewers previous suggestions: adding in discussion of the connection between cardiac disease, GJC, and diabetes.  Although this was not addressed it is not of major concern and could be the focus of future review from the authoring group.

Author Response

Thank you very much for your review and for your comments. They were a great contribution to our work that made it more valuable.
